

# Potential therapeutic agents of *Bombyx mori* silk cocoon extracts from agricultural product for inhibition of skin pathogenic bacteria and free radicals

Thida Kaewkod[1,2], Puangphaka Kumseewai[1], Sureeporn Suriyaprom[1,3], Varachaya Intachaisri[1], Nitsanat Cheepchirasuk[1] and Yingmanee Tragoolpua[1,2]

[1] Department of Biology, Faculty of Science, Chiang Mai University, Chiang Mai, Thailand
[2] Natural Extracts and Innovative Products for Alternative Healthcare Research Group, Chiang Mai University, Chiang Mai, Thailand
[3] Office of Research Administration, Chiang Mai University, Chiang Mai, Thailand

Corresponding author
Yingmanee Tragoolpua,
yboony150@gmail.com,
yingmanee.t@cmu.ac.th

## ABSTRACT

**Background**. Pathogenic bacteria are the cause of most skin diseases, but issues such as resistance and environmental degradation drive the need to research alternative treatments. It is reported that silk cocoon extract possesses antioxidant properties. During silk processing, the degumming of silk cocoons creates a byproduct that contains natural active substances. These substances were found to have inhibitory effects on bacterial growth, DNA synthesis, the pathogenesis of hemolysis, and biofilm formation. Thus, silk cocoon extracts can be used in therapeutic applications for the prevention and treatment of skin pathogenic bacterial infections.

**Methods**. The extract of silk cocoons with pupae (SCP) and silk cocoons without pupae (SCWP) were obtained by boiling with distilled water for 9 h and 12 h, and were compared to silkworm pupae (SP) extract that was boiled for 1 h. The active compounds in the extracts, including gallic acid and quercetin, were determined using high-performance liquid chromatography (HPLC). Furthermore, the total phenolic and flavonoid content in the extracts were investigated using the Folin–Ciocalteu method and the aluminum chloride colorimetric method, respectively. To assess antioxidant activity, the extracts were evaluated using the 2,2-diphenyl–1-picrylhydrazyl (DPPH) radical scavenging assay. Additionally, the minimal inhibitory concentration (MIC) and minimal bactericidal concentration (MBC) of silk extracts and phytochemical compounds were determined against skin pathogenic bacteria. This study assessed the effects of the extracts and phytochemical compounds on growth inhibition, biofilm formation, hemolysis protection, and DNA synthesis of bacteria.

**Results**. The HPLC characterization of the silk extracts showed gallic acid levels to be the highest, especially in SCP (8.638–31.605 mg/g extract) and SP (64.530 mg/g extract); whereas quercetin compound was only detected in SCWP (0.021–0.031 mg/g extract). The total phenolics and flavonoids in silk extracts exhibited antioxidant and antimicrobial activity. Additionally, SCP at 9 h and 12 h revealed the highest anti-bacterial activity, with the lowest MIC and MBC of 50–100 mg/mL against skin pathogenic bacteria including *Staphylococcus aureus*, methicillin-resistant *S. aureus*

(MRSA), *Cutibacterium acnes* and *Pseudomonas aeruginosa*. Hence, SCP extract and non-sericin compounds containing gallic acid and quercetin exhibited the strongest inhibition of both growth and DNA synthesis on skin pathogenic bacteria. The suppression of bacterial pathogenesis, including preformed and matured biofilms, and hemolysis activity, were also revealed in SCP extract and non-sericin compounds. The results show that the byproduct of silk processing can serve as an alternative source of natural phenolic and flavonoid antioxidants that can be used in therapeutic applications for the prevention and treatment of pathogenic bacterial skin infections.

# INTRODUCTION

The process of cultivating silkworms and extracting silk, known as sericulture, is a part of Southeast Asia's cultural heritage. Sericulture in Thailand was established about one hundred years ago, and today the silk industry continues to provide valuable and important export products (*Sramala, 2018*). Silk cocoons are produced by silkworms; *Bombyx mori* L., by generating twin silk filaments and wrapping them around the body of the silkworm (*Devi et al., 2011*). Extracting the silk fibers involves degumming the silk cocoon, which is performed by boiling cocoons in water for more than one hour (*Javali et al., 2015*). This water is found to contain substances from the *B. mori* cocoon shell and silkworm. The *B. mori* cocoon shell consists of silk fibroin fiber (70%), sericin (25%) and non-sericin (5%). Sericin has many beneficial biological properties, including anti-cancer, antioxidant, skin whitening and UV-resistant properties (*Sasaki, Kato & Watanabe, 2000*; *Dash et al., 2008*). The non-sericin components are mainly comprised of flavonoid compounds, found in the adjacent layers of the sericin protein within the silkworm cocoon, and their presence imparts a yellow color to the silk. Other non-sericin components include antimicrobial proteins including seroins and protease inhibitors (*Dong et al., 2016*). In addition to flavonoids and carotenoids, there are also various non-protein components in silkworm cocoons, including organic acids, fatty acids, carbohydrates, amino acids, and hydrocarbons (*Zhang et al., 2017*). The color of the cocoon is determined by two primary pigment categories; yellowish carotenoids, which are soluble in ether, and green flavonoids, such as quercetin 5-glucoside, quercetin 5,40-diglucoside, and quercetin 5,7, 4′-triglucoside, which are soluble in water (*Tamura et al., 2002*). Moreover, the color constituents are linked to the phenolic compounds found in the leaves of mulberries (*Morus alba* L.), the exclusive diet of silkworm larvae (*Kurioka & Yamazaki, 2002*). Natural phenolics and flavonoids are effective antioxidants that scavenge oxygen radicals and possess anti-aging, hypolipidemic, anticancer and anti-inflammatory properties (*Yang et al., 2012*). However, the flavonoid and phenolic compounds obtained from the degumming process have not yet been widely studied. Therefore, this study aims to investigate the compounds in silk cocoon extract in

order to understand the phytochemical composition, and the antioxidant and antibacterial activity.

Skin covers most of the human body and acts as the body's primary barrier to the environment, helping to protect against physical impacts and prevent dehydration. Notably, the skin plays a crucial role in safeguarding the body from infections by preventing the entry of invading pathogens (*Proksch, Brandner & Jensen, 2008*). The cutaneous defense mechanisms encompass the firmness and low moisture content of the stratum corneum. Additionally, the skin's outer layer produces stratum corneum lipids, lysozyme, acidity (pH 5), and defensins (*Menon, Cleary & Lane, 2012*). Although the skin functions as protection from harmful microbes, it also provides an area of about 30 square meters of diverse microbial habitat (*Gallo, 2017*). When a person has a wound or a weak immune system, the risk of a skin infection increases. Skin diseases globally, especially bacterial skin infections, have increased marginally by 7.38% from 1990 to 2019 (*Xue et al., 2022*). In Thailand, infections of the skin and subcutaneous tissues recorded between 2015-2019 were the most common (37.3%), followed by dermatitis (29.7%), urticaria and erythema (13.9%), other disorders of the skin and subcutaneous tissue (8.6%), and papulosquamous disorders (1.7%) (*Chaiyamahapurk & Warnnissorn, 2021*).

Pathogenic bacteria like *Staphylococcus aureus*, *Pseudomonas aeruginosa*, and *Cutibacterium acnes* are common culprits in causing skin infections (*Grice & Segre, 2011*). A high virulence strain of methicillin-resistant *S. aureus* (MRSA), for example, has become resistant to methicillin and other similar antibiotics such as penicillin and vancomycin (*Chambers, 2003*). Moreover, the normal skin flora, which includes *Staphylococcus epidermidis, Micrococcus luteus, Bacillus* sp., and *Corynebacterium* sp., can potentially lead to opportunistic infections in individuals with wounds or compromised immunity. The process of a skin infection begins when bacteria attach to the cells of a hair follicle, multiply, and spread downward to the follicle and sebaceous glands (*Rosenthal et al., 2011*). Notably, Staphylococcal bacteria could potentially produce the hemolysin toxin, which is an important virulence factor that could damage red blood cells, causing lysis and eventually cell death. The actions of hemolysin lead to bacterial invasion and evade the response of the host immune system. (*Rajagopal, 2009*). *Pseudomonas aeruginosa* generates adhesion factors, exotoxin A, phospholipase C for hemolysis, and exoenzyme S, contributing to its role in acute infections (*Khalifa et al., 2011*). While drugs and synthetic agents are frequently used in the treatment of bacterial infections, resistance to certain medications has emerged. Consequently, there is now active research exploring bioactive compounds, such as silk cocoon and silkworm extracts, as beneficial alternatives providing safe and effective treatment of human skin infections. In the present study, we have evaluated the effects of silk cocoon and silkworm extract on the growth of skin pathogenic bacteria. The effects on preformed and matured biofilms, hemolysis protection and DNA synthesis of the bacteria were also studied, as well as the antioxidant and phytochemical compounds present in these extracts.

## MATERIALS & METHODS

### Preparation of silk cocoon and silkworm pupae extracts

Silk cocoons and silkworm pupae were provided by Piankusol Silk and Cotton Co., Ltd., Chiang Mai, Thailand. The extracts of silk cocoons with pupae and silk cocoons without pupae were boiled in sterile, distilled water at 100 °C for 9 h and 12 h. Silkworm pupae extract was boiled in sterile, distilled water at 100 °C for 1 h (*Jantakee et al., 2021*). The sample and solvent were used at a ratio of 1:25 (weight per volume). After that, the extract was filtered using Whatman No.1 filter paper (GE Healthcare UK Limited, Shanghai, China). The filtrate of the extract was then evaporated using a rotary evaporator (Heidolph, Schwabach, Germany) at 45 °C under reduced pressure at 50 mbar, and freeze dried using a lyophilizer (LABCONCO, Kansas City, MO, USA) at −50 °C under reduced pressure. The crude extract was dissolved in sterile, distilled water at a concentration of 200 mg/mL.

### Total phenolic content

The total phenolic content in silk cocoon and silkworm pupae extracts was investigated using the Folin–Ciocalteu method (*Singhatong, Leelarungrayub & Chaiyasut, 2010*). The extract (250 µL) was mixed with 125 µL of 50% Folin–Ciocalteu reagent (Merck, Darmstadt, Germany) and 250 µL of 95% ethanol (RCI Labscan Limited, Bangkok, Thailand). The mixture was kept in darkness at room temperature (25 °C) for 5 min. Following incubation, 250 µL of 5% sodium carbonate was introduced, and the mixture was then kept in darkness at room temperature (25 °C) for 60 min. The formation of a blue molybdenum-tungsten complex occurred prior to detection at an absorbance of 725 nm. The total phenolic content was calculated by comparing the samples to the content of standard gallic acid, and is expressed as milligrams of gallic acid equivalent per gram of extract (mg GAE/g extract).

### Total flavonoid content

The total flavonoid content was determined using the aluminum chloride colorimetric method (*Deori et al., 2014*). The extract was two-fold serial diluted with methanol. The reaction mixture comprised 0.5 mL of extract, 0.1 mL of 10% aluminum chloride, 1.5 mL of methanol, 0.1 mL of potassium acetate, and 2.8 mL of distilled water. The reaction was left to incubate in darkness at room temperature (25 °C) for 30 min. Subsequently, the absorbance was gauged at a wavelength of 415 nm. The total flavonoid content was measured by comparing the samples to standard quercetin, and is expressed as milligrams of quercetin equivalent per gram of extract (mg QUE/g extract).

### Detection of gallic acid and quercetin using high-performance liquid chromatography (HPLC)

High-performance liquid chromatography (HPLC) was employed to ascertain the levels of gallic acid and quercetin in extracts from silk cocoons and silkworm pupae. The extracts, along with the standard compounds of gallic acid (Sigma-Aldrich, Darmstadt, Germany) and quercetin (Sigma-Aldrich), underwent filtration through a 0.45 µm sterile microfilter. Subsequently, 20 µL of the filtered sample was injected into the HPLC system (Agilent

Technologies 1,200 series; Agilent Technologies, Santa Clara, CA, USA). Further analysis was conducted using a 320 nm UV detector by GL Sciences C-18 column (4.6 × 150 mm, 5 μm, Tokyo, Japan). The mobile phase, utilized for separation, was a combination of mobile phase A (1% formic acid in water) and mobile phase B (acetonitrile). Gradient elution was performed at various times with different ratios of mobile phase A and B as follows; ratio of 90:10 (0 min), ratio of 80:20 (5 min), ratio of 75:25 (10 min), ratio of 70:30 (15 min), ratio of 65:35 (20 min), ratio of 60:40 (25 min) and ratio of 50:50 (30 min). The HPLC condition was used at a flow rate of 1.0 ml/min and a running time of 30 min (*Zhou et al., 2013*). The amount of gallic acid and quercetin in the extracts was calculated in comparison to the standard for each compound.

## Determination of antioxidant activity of silk cocoon and silkworm pupae extracts

The antioxidant activity of the extracts was evaluated by 2,2 diphenyl-1-picrylhydrazyl (DPPH) radical scavenging assay (*Prior, Wu & Schaich, 2005*; *Ghasemi, Ghasemi & Ebrahimzadeh, 2009*). The extracts were prepared with methanol at various concentrations. The reaction involved combining 0.5 mL of the extract with 1.5 mL of DPPH solution, followed by incubation in darkness at room temperature (25 °C) for 20 min. The absorbance was then measured at a wavelength of 517 nm. A control was established using DPPH without extract, and a blank solution was prepared using methanol. The absorbance of the DPPH solution A1, and the absorbance of the extracts with the DPPH solution A2, were measured. The percentage of DPPH free radical inhibition was calculated as follows: percentage inhibition = {(A1–A2)/A1} × 100.

The antioxidant activity of the extracts was assessed by comparing the samples to standard gallic acid, and is expressed as milligrams of gallic acid equivalent per gram of extract (mg GAE/g extract).

## Bacterial strains

The skin pathogenic bacteria *Staphylococcus aureus* ATCC 25923, methicillin resistant *S. aureus* (MRSA), *Cutibacterium acnes* DMST 14916 and *Pseudomonas aeruginosa* ATCC 27853 were used in this study. The bacteria were obtained from the Division of Microbiology, Department of Biology, Faculty of Science, Chiang Mai University, Chiang Mai, Thailand. The bacteria were cultured in Mueller Hinton (MH) broth after incubation under aerobic conditions at 37 °C for 24 h, except *C. acnes,* which was cultured in Brain Heart infusion (BHI) after incubation under anaerobic conditions at 37 °C for 72 h.

## Determination of minimal inhibitory concentration (MIC) and minimal bactericidal concentration (MBC) of silk extracts and phytochemical compounds on skin pathogenic bacteria

Minimal inhibitory concentration (MIC) was determined by serial tube dilution technique. Different concentrations of the extracts and phytochemical compounds were two-fold serial diluted with MHB medium. Subsequently, the bacterial culture was adjusted to McFarland No. 0.5 ($10^8$ CFU/mL) before adding it to each dilution of extracts at a ratio of 1:1. The

MHA agar plate was incubated at 37 °C for 18–24 h under aerobic conditions, except *C. acnes,* which was incubated at 37 °C for 72 h under anaerobic conditions. The gentamycin antibiotic was used as a positive control. Following incubation, the lowest inhibitory concentration of the extract, where no bacterial growth was observed comparing the control, was recorded as turbidity. The culture broth exhibiting no discernible bacterial growth was streaked on an MHA agar plate, and the presence of bacterial colonies was ascertained after further incubation. The MBC value was determined from the concentration that inhibited bacterial growth by 99.9%, following the methodology outlined by *Andrews (2001)*.

### Determination of time-kill curve assay of silk cocoon and silkworm pupae extracts on bacterial growth

The bacterial culture was standardized to McFarland standard No. 0.5 and blended with the crude extracts at the concentration corresponding to the MIC in a 1:1 ratio. This bacterial mixture underwent incubation at 37 °C for 24 h, with samples collected at intervals of 0, 2, 4, 6, 8, 10, 12, and 24 h. The bacteria were ten-fold serial diluted and spread on agar medium. After incubation, the bacterial colonies were counted and the number of bacteria was calculated as colony forming units (CFU/mL). The time-kill curve of bacteria was compared to the number of bacteria treated with extract, and the number of bacteria without extract (*Olajuyigbe & Afolayan, 2012*).

### Effect of silk cocoon and silkworm pupae extracts and phytochemical compounds on preformed biofilm

The effect of silk extracts and phytochemical compounds on air-liquid interface biofilm was determined by biofilm assay (*Mathur et al., 2006*). The 96-well polystyrene plates contained 100 μL of bacterial culture ($10^8$ CFU/mL) in the absence and presence of silk cocoon extract, silkworm pupae extract and phytochemical compounds. Following a 24 h incubation period at 37 °C, the planktonic cells in the wells were eliminated and subjected to two washes with sterile distilled water. Subsequently, the wells were stained with crystal violet (0.4%) for 20 min, followed by two additional washes with sterile distilled water before air drying. The crystal violet was resuspended using 95% ethanol and the absorbance was measured at a wavelength of 592 nm. The percentage of biofilm inhibition was calculated in comparison to the control cells.

### Effect of silk cocoon and silkworm pupae extracts and phytochemical compounds on mature biofilm

The effect of silk extracts and phytochemical compounds on mature biofilm was assessed by following the technique of *Mathur et al. (2006)*, with some modifications. Bacterial culture ($10^8$ CFU/mL) was grown on 96-well polystyrene plates for 24 h. The free cells in the wells were removed and washed twice with phosphate buffer saline (PBS, pH 7.4). Each concentration of silk cocoon and silkworm pupae extracts, and phytochemical compounds were added to each well. After incubation at 37 °C for 24 h, the sample solution was discarded and washed twice with sterile distilled water. The wells were then stained with crystal violet (0.4%) for 20 min and washed twice with sterile distilled water before air drying. The crystal violet was resuspended using 95% ethanol and the absorbance was

measured at a wavelength of 592 nm. The percentage of mature biofilm inhibition was calculated in comparison to the control cells.

### Effect of silk cocoon and silkworm pupae extracts and phytochemical compounds on hemolysis protection

The effect of silk cocoon and silkworm pupae extracts and their phytochemical compounds on hemolysis protection from bacterial infection was performed using hemolysis quantification assay (*Kannappan et al., 2017*). The bacterial culture ($10^8$ CFU/mL) was incubated with silk cocoon and silkworm pupae extracts and phytochemical compounds for 18–24 h under a 37 °C incubator. To assess hemolytic activity, the tested bacteria were incubated with an equal volume of red blood cells (2% v/v in PBS) for 1 h at 37 °C. Following incubation, the tubes underwent centrifugation at 3,000 g for 10 min, and the absorbance of the supernatant was measured at a wavelength of 405 nm. The percentage of hemolysin inhibition was calculated in comparison to the control cells.

### Effect of silk cocoon and silkworm pupae extracts and phytochemical compounds on bacterial DNA synthesis

The DNA preparation used in this study follows the technique of *Siripornmongcolchai et al. (2002)*. The bacteria were standardized to McFarland No. 0.5 and subjected to treatment with the extracts. The combination of bacterial culture and extract underwent incubation at 37 °C for 24 h under aerobic conditions, except for *C. acnes*, which was incubated at 37 °C for 72 h under anaerobic conditions. Bacterial cells were collected after centrifugation at 6,000 rpm at 4 °C for 10 min, then reconstituted with 560 µL of TE buffer (10 mM Tris, pH 8; 1 mM EDTA, pH 8). The lysis solution, comprising 10% sodium dodecyl sulfate (SDS), was incorporated with proteinase K solution (20 mg/mL) before incubation at 37 °C for 1 h. Additionally, 100 µL of 5 M sodium chloride was introduced to the mixture, followed by incubation at 65 °C for 10 min. DNA extraction was carried out using phenol-chloroform, and precipitation was induced by 600 µL of ice-cold absolute ethanol. The DNA pellet was washed with 70% ethanol and resuspended in 50 µL of TE buffer. Following treatment with extracts, the DNA quantity was determined using BioDrop (TOUCH UV/Visible Spectrophotometer; ISS, San Antonio, TX, USA), and the results were compared to the DNA of the untreated control. The percentage of inhibition of DNA synthesis was subsequently calculated.

### Statistical analysis

All experiments were performed in three independent treatments. All data from the treatments and the control groups were compared, analyzed and presented as mean ± SD using $t$-test and ANOVA analysis.

## RESULTS

### Total phenolic and flavonoid content in silk cocoon and silkworm pupae extracts

The total phenolic and flavonoid content in silk cocoon and silkworm pupae extracts are presented in Table 1. The highest total phenolic compound was significantly visible in the

**Table 1 Total amount of phenolic and flavonoid content in silk cocoon and silkworm pupae extracts.**

| Extracts | Extraction method | Phenolics content (mg GAE/g extract) | Flavonoids content (mg QUE/g extract) |
|---|---|---|---|
| SCP | Boiling for 9 h | $13.54 \pm 0.34^a$ | $1.31 \pm 0.37^a$ |
| | Boiling for 12 h | $6.90 \pm 0.09^b$ | $1.58 \pm 0.12^a$ |
| SCWP | Boiling for 9 h | $4.93 \pm 0.22^a$ | $0.79 \pm 0.23^a$ |
| | Boiling for 12 h | $5.98 \pm 0.20^a$ | $0.80 \pm 0.00^a$ |
| SP | Boiling for 1 h | $16.04 \pm 0.46^*$ | $2.14 \pm 0.01^*$ |

Notes.

The data of different superscript letters (a, b) are presented as mean $\pm$ SD of triplicate independent experiments and show significantly different values in each extract ($P < 0.05$). All data are used to analyze between two groups using one-way ANOVA and Tukey's test for multiple comparisons.

*Values show the highest significant of each compound ($P < 0.05$).

Silk cocoon with pupae (SCP), silk cocoon without pupae (SCWP) and silkworm pupae (SP).

silkworm pupae (SP) extract after boiling for 1 h, with a value of $16.04 \pm 0.46$ mg GAE/g extract. Moreover, the extract from silk cocoons with pupae (SCP) had a higher total phenolic content than the extract from silk cocoons without pupae (SCWP) ($P < 0.05$). Particularly, SCP extracted by boiling for 9 h had a phenolic content of $13.54 \pm 0.34$ mg GAE/g extract, presenting a higher content than SCP extracted by boiling for 12 h (Table 1).

Extracts from SP after boiling for 1 h significantly showed the highest total flavonoid compound of $2.14 \pm 0.01$ mg QUE/g extract. The extract of silk cocoons with the presence and absence of pupae after boiling for 12 h showed a higher total flavonoid content than the extraction boiled for 9 h. The total flavonoid content in SCP after boiling for 9 h and 12 h was $1.31 \pm 0.37$ and $1.58 \pm 0.12$ mg QUE/g extract, respectively. In addition, the total flavonoid content in SCWP after boiling for 9 h and 12 h was $0.79 \pm 0.23$ and $0.80 \pm 0.00$ mg QUE/g extract, respectively. Therefore, SCP after boiling demonstrated a higher flavonoid content than SCWP (Table 1).

## Gallic and quercetin content in silk cocoon and silkworm pupae extracts

The phytochemical compounds of gallic acid and quercetin, which are present in silk cocoon and silkworm pupae extracts, were measured by HPLC assay. The results revealed that SCWP extract contained both gallic acid and quercetin after boiling for 9 and 12 h (Table 2 and Fig. 1). The highest content of gallic acid and quercetin was found in SP and SCWP extracts (12 h), showing significant values of $64.530 \pm 0.120$ mg/g extract and $0.031 \pm 0.000$ mg/g extract, respectively. Additionally, the highest content of gallic acid was found in silk cocoon extract after boiling for 9 h. In contrast, quercetin content in SCWP after boiling for 12 h was found to be significantly higher than the extract boiled for 9 h ($P < 0.05$) (Table 2).

## Antioxidant activity of silk cocoon and silkworm pupae extracts

The DPPH radical scavenging activities of silk extracts are presented in Table 3. Results show that SP extract demonstrated the strongest antioxidant activity of $14.03 \pm 1.49$ mg GAE/g extract. SCP and SCWP extracted by boiling for 12 h displayed higher antioxidant

**Table 2** The amount of gallic acid and quercetin in silk cocoon and silkworm pupae extracts.

| Extracts | Extraction method | Gallic acid (mg/g extract) | Quercetin (mg/g extract) |
|---|---|---|---|
| SCP | Boiling for 9 h | $31.605 \pm 0.015^a$ | Not detected |
| | Boiling for 12 h | $8.638 \pm 0.038^b$ | Not detected |
| SCWP | Boiling for 9 h | $9.688 \pm 0.040^a$ | $0.021 \pm 0.001^a$ |
| | Boiling for 12 h | $8.608 \pm 0.062^b$ | $0.031 \pm 0.000^{b,*}$ |
| SP | Boiling for 1 h | $64.530 \pm 0.120^*$ | Not detected |

Notes.

The data of different superscript letters (a, b) are presented as mean $\pm$ SD of triplicate independent experiments and show significantly different values in each extract ($P < 0.05$). All data are used to analyze between two groups using one-way ANOVA and Tukey's test for multiple comparisons.

*Values show the significant highest of each compound ($P < 0.05$).

Silk cocoon with pupae (SCP), silk cocoon without pupae (SCWP) and silkworm pupae (SP).

activity than the extract obtained by boiling for 9 h. The antioxidant activities of SCP and SCWP after boiling for 12 h were $10.96 \pm 2.35$ mg GAE/g extract and $3.22 \pm 0.94$ mg GAE/g extract, respectively (Table 3).

## Antibacterial activity of silk cocoon and silkworm pupae extracts, and phytochemical compounds

The antibacterial activity of silk extracts and the phytochemical compounds of gallic acid and quercetin was also measured against the skin pathogenic bacteria *S. aureus*, methicillin resistant *S. aureus* (MRSA), *C. acnes* and *P. aeruginosa*. The MIC and MBC values were determined by broth dilution method. The results showed that SCP and SCWP and the phytochemical compounds of gallic acid and quercetin had the ability to inhibit skin pathogenic bacteria (Table 4). The SCP extracted by boiling for 9 and 12 h showed the lowest MIC and MBC values of 50 mg/mL against MRSA, *C. acnes* and *P. aeruginosa*. Moreover, the lowest MIC and MBC values of 50 mg/mL inhibited *C. acnes* from SCWP extracted by boiling for 12 h. In addition, SP extract displayed the lowest MIC and MBC values of 25 mg/mL against *C. acnes* (Table 4). The phytochemical compounds of gallic acid and quercetin in silk extracts also inhibited skin pathogenic bacteria, with the lowest MIC and MBC values ranging from 1.25–5.0 mg/mL (Table 4).

## Time-kill kinetics of silk cocoon with pupae extracts on bacterial growth

As silk cocoon with pupae extract (SCP) obtained by boiling for 9 and 12 h showed the greatest antibacterial activity on skin pathogenic bacteria, the time-kill kinetic profiles of these extracts were tested on *S. aureus*, MRSA and *P. aeruginosa* at MIC concentrations. In this study, MIC inhibitory concentrations of SCP obtained by boiling for 9 h and 12 h were used against MRSA, *P. aeruginosa* (50 mg/mL) and *S. aureus* (100 mg/mL). The results are presented in terms of the changes in the $\log_{10}$ reduction CFU per ml of viable colonies, and the extract was found to exhibit significant bactericidal activity (Fig. 2). After incubating SCP obtained by boiling for 9 h with *S. aureus*, MRSA and *P. aeruginosa*, the average log reduction of the viable cells showed complete inhibition within 6, 10 and 4 h, respectively. For the time-kill kinetics of SCP obtained by boiling for 12 h, the average $\log_{10}$ reduction

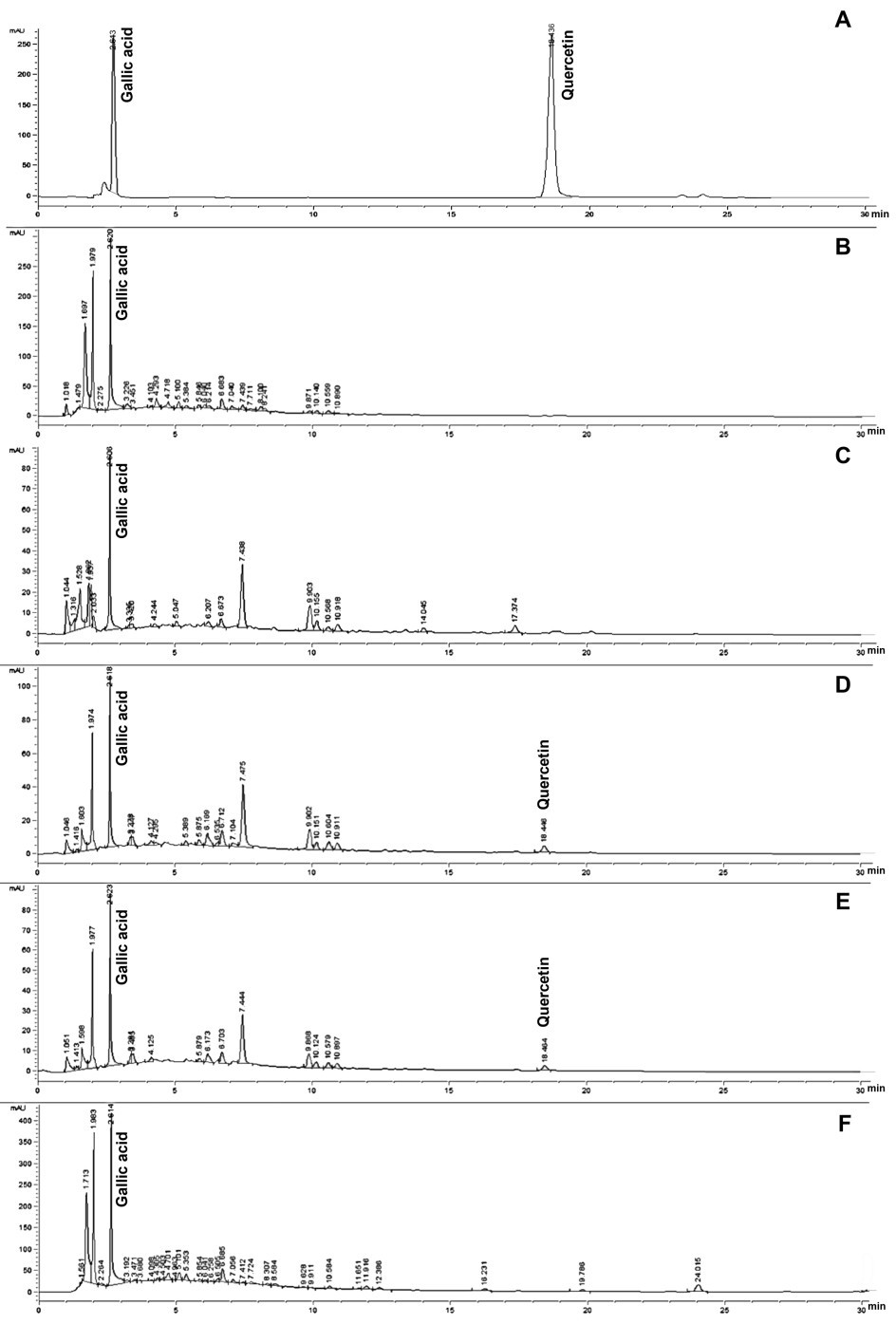

**Figure 1** HPLC chromatograms of gallic acid and quercetin (A) founded in SCP extracted by boiling for 9 h and 12 h (B and C), SCWP extracted by boiling for 9 h and 12 h (D and E) and SP extracted by boiling for 1 h (F) extracts.

**Table 3** The antioxidant activity of silk cocoon and silkworm pupae extracts by DPPH radical scavenging assay.

| Extracts | Extraction method | Antioxidant activity (mg GAE/g extract) |
|---|---|---|
| SCP | Boiling for 9 h | 9.61 ± 1.20[a] |
| | Boiling for 12 h | 10.96 ± 2.35[a] |
| SCWP | Boiling for 9 h | 2.76 ± 1.67[a] |
| | Boiling for 12 h | 3.22 ± 0.94[a] |
| SP | Boiling for 1 h | 14.03 ± 1.49[*] |

Notes.

The data of different superscript letters (a, b) are presented as mean ± SD of triplicate independent experiments and show significantly different values in each extraction method ($P < 0.05$). All data are used to analyze between two groups using one-way ANOVA and Tukey's test for multiple comparisons.

*Values show the highest significant antioxidant activity ($P < 0.05$).

Silk cocoon with pupae (SCP), silk cocoon without pupae (SCWP) and silkworm pupae (SP).

**Table 4** Minimal inhibitory concentration (MIC) and minimal bactericidal concentration (MBC) values of silk cocoon and silkworm pupae extracts against skin pathogenic bacteria using broth dilution method.

| Samples | MIC and MBC (mg/mL) | | | | | | | |
|---|---|---|---|---|---|---|---|---|
| | *S. aureus* | | MRSA | | *C. acnes* | | *P. aeruginosa* | |
| | MIC | MBC | MIC | MBC | MIC | MBC | MIC | MBC |
| Silk extracts | | | | | | | | |
| SCP (Boiling for 9 h) | 100 ± 0 | 100 ± 0 | 50 ± 0 | 50 ± 0 | 50 ± 0 | 50 ± 0 | 50 ± 0 | 50 ± 0 |
| SCP (Boiling for 12 h) | 100 ± 0 | 100 ± 0 | 50 ± 0 | 50 ± 0 | 50 ± 0 | 50 ± 0 | 50 ± 0 | 50 ± 0 |
| SCWP (Boiling for 9 h) | 100 ± 0 | 100 ± 0 | 100 ± 0 | 100 ± 0 | 100 ± 0 | 100 ± 0 | 100 ± 0 | 100 ± 0 |
| SCWP (Boiling for 12 h) | 100 ± 0 | 100 ± 0 | 100 ± 0 | 100 ± 0 | 50 ± 0 | 50 ± 0 | 100 ± 0 | 100 ± 0 |
| SP (Boiling for 1 h) | 100 ± 0 | 100 ± 0 | 100 ± 0 | 100 ± 0 | 25 ± 0 | 25 ± 0 | 100 ± 0 | 100 ± 0 |
| Phytochemical compounds | | | | | | | | |
| Gallic acid | 2.5 ± 0 | 2.5 ± 0 | 5.0 ± 0 | 5.0 ± 0 | 1.25 ± 0 | 1.25 ± 0 | 2.5 ± 0 | 2.5 ± 0 |
| Quercetin | 2.5 ± 0 | 2.5 ± 0 | 5.0 ± 0 | 5.0 ± 0 | 1.25 ± 0 | 1.25 ± 0 | 2.5 ± 0 | 2.5 ± 0 |
| Antibiotic control | | | | | | | | |
| Gentamycin | 0.04 ± 0 | 0.04 ± 0 | 0.25 ± 0 | 0.25 ± 0 | 12.5 ± 0 | 12.5 ± 0 | 12.5 ± 0 | 12.5 ± 0 |

Notes.

The results are presented as mean ± SD of three independent experiments.

Silk cocoon with pupae (SCP), silk cocoon without pupae (SCWP) and silkworm pupae (SP).

in the viable cells was completely inhibited within 2 h after MRSA and *P. aeruginosa* were incubated with these extracts (Fig. 2).

## Effect of silk cocoon with pupae extracts and phytochemical compounds on preformed biofilm

The inhibition of preformed biofilm on skin pathogenic bacteria was investigated by crystal violet biofilm assay. After bacteria were incubated with sub-MIC concentrations of SCP and phytochemical compounds, SCP obtained by boiling for 12 h (SCP at 12 h) at a concentration of 25 mg/mL had the ability to inhibit biofilm formation of *S. aureus* and *C. acnes*, and demonstrated significantly higher inhibitory activity than SCP obtained by boiling for 9 h (SCP at 9 h), by 72.53% and 66.67%, respectively (Fig. 3). In addition, SCP

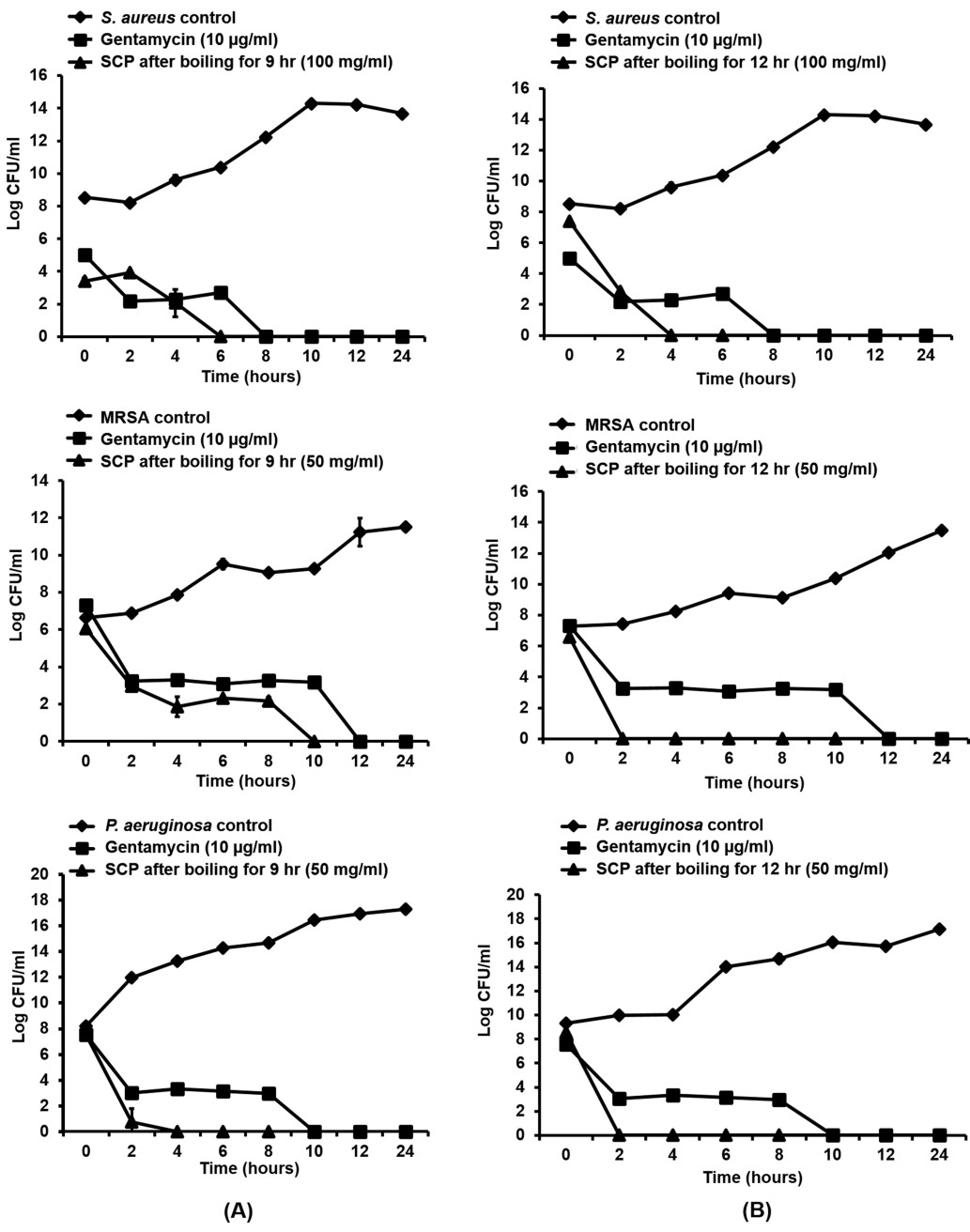

**Figure 2 Time-kill kinetics of silk cocoon with pupae (SCP) extracted by boiling for 9 h (A) and 12 h (B) against *S. aureus*, MRSA and *P. aeruginosa*.** The average log reduction of viable bacteria cells at each time point is presented as mean ± SD of three independent experiments.

at 9 h (25 mg/mL) demonstrated a larger reduction in biofilm formation of MRSA and *P. aeruginosa* than SCP at 12 h, with inhibition values of 93.14% and 79.18%, respectively (Fig. 3). Moreover, the phytochemical compounds of gallic acid and quercetin at sub-MIC concentrations (1.25–2.5 mg/mL) inhibited biofilm formation of *S. aureus*, MRSA and *P. aeruginosa* with reduction values of more than 90%. In contrast, the biofilm formation

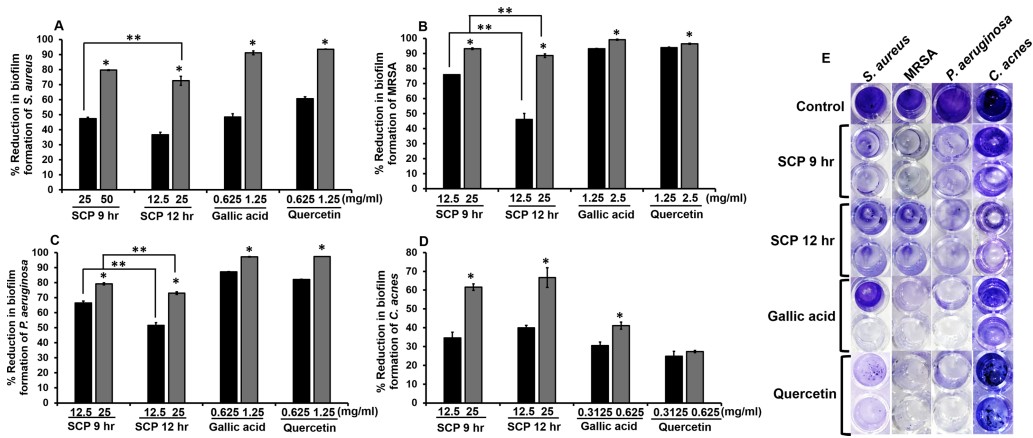

**Figure 3** **Effect of silk cocoon with pupae extracts from boiling for 9 h (SCP 9 h) and 12 h (SCP 12 h) and phytochemical compounds of gallic acid and quercetin on preformed biofilm of *S. aureus* (A), MRSA (B), *P. aeruginosa* (C) and *C. acnes* (D) after staining with crystal violet (E).** The average percentage of biofilm reduction of bacteria is presented as mean ± SD of three independent experiments. *Values show the highest significant biofilm inhibition ($P < 0.05$). **Biofilm inhibition shows significantly different values between SCP 9 h and SCP 12 h ($P < 0.05$).

of *C. acnes* was inhibited by gallic acid and quercetin at a concentration of 0.625 mg/mL by 41.12% and 27.35%, respectively (Fig. 3).

## Effect of silk cocoon with pupae extracts and phytochemical compounds on mature biofilm

The degradation of mature biofilm on skin pathogenic bacteria was investigated by crystal violet biofilm assay. After bacteria were incubated with sub-MIC concentrations of SCP and phytochemical compounds, SCP obtained by boiling for 12 h (SCP at 12 h) at a concentration of 25 mg/mL had the ability to degrade mature biofilm of *P. aeruginosa* and *C. acnes*, and showed significantly higher reduction activity than SCP from boiling for 9 h (SCP at 9 h) by 92.81% and 91.34%, respectively (Fig. 4). Additionally, SCP at 9 h (25 mg/mL) demonstrated degradation of mature biofilm on MRSA and *P. aeruginosa* with reduction values of 87.56% and 84.84%, respectively (Fig. 4). The phytochemical compounds of gallic acid and quercetin at sub-MIC concentrations (1.25–2.5 mg/mL) degraded mature biofilm of *S. aureus*, MRSA and *P. aeruginosa* with reduction values of 40.38–92.62%. Moreover, the mature biofilm of *C. acnes* was degraded by gallic acid and quercetin at concentrations of 0.3125–0.625 mg/mL with reduction values of 53.33–88.38% (Fig. 4).

## Effect of silk cocoon with pupae extracts and phytochemical compounds on hemolysis protection

The extracellular protein of hemolysin from bacteria causes hemolysis by the breakdown or destruction of red blood cells. The inhibition of hemolysis by SCP extracts and phytochemical compounds on skin pathogenic bacteria were studied by observing hemolysis activity. The results showed that SCP at 9 h and SCP at 12 h at the concentration

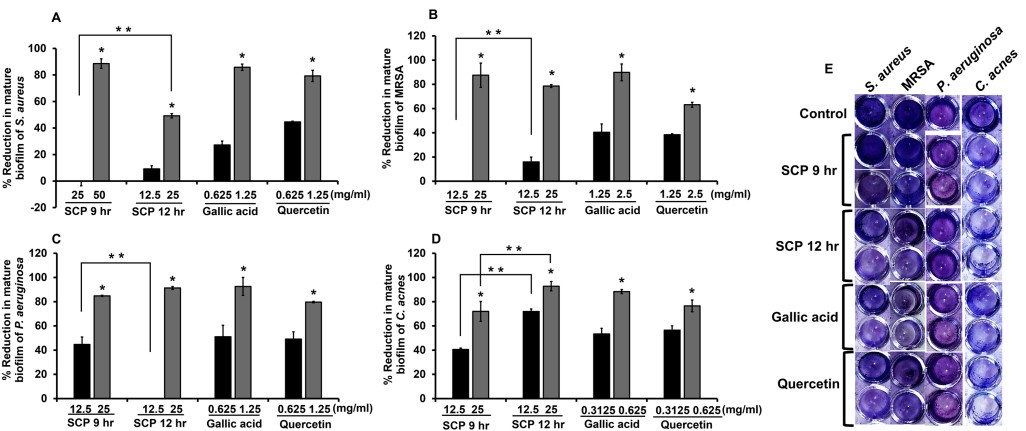

**Figure 4** Effect of silk cocoon with pupae extracts from boiling for 9 h (SCP 9 h) and 12 h (SCP 12 h) and phytochemical compounds of gallic acid and quercetin on the mature biofilm of *S. aureus* (A), MRSA (B), *P. aeruginosa* (C) and *C. acnes* (D) after staining with crystal violet (E). The average percentage of mature biofilm reduction of bacteria is presented as mean ± SD of three independent experiments. [*]Values show the highest significant mature biofilm inhibition ($P < 0.05$). [**]Biofilm degradation shows significantly different values between SCP 9 h and SCP 12 h ($P < 0.05$).

of 25 mg/mL could prevent hemolysis by more than 60% in *S. aureus*, MRSA, *P. aeruginosa* and *C. acnes* infections (Fig. 5). SCP at 12 h had the ability to prevent hemolysis by 91.44–95.38% in *S. aureus*, MRSA, and *P. aeruginosa* infections, which is greater than SCP at 9 h, with inhibitory activity in the ranges of 60.71–94.12%. In contrast, hemolysis activity in *C. acnes* exhibited 82.21% inhibition by SCP at 9 h (25 mg/mL), which is higher than SCP at 12 h, with 68.83% inhibition (Fig. 5). Moreover, the phytochemical compound of gallic acid at sub-MIC concentrations was able to prevent hemolysis in all tested bacteria by 78.54–93.86%. Additionally, quercetin was able to prevent bacterial hemolysis by 61.56–74.38% (Fig. 5).

## Effect of silk cocoon with pupae extracts and phytochemical compounds on bacterial DNA synthesis

Silk cocoon with pupae extracts (SCP) and phytochemical compounds were studied for their ability to inhibit bacterial DNA synthesis. SCP at 9 h had the ability to inhibit bacterial DNA synthesis in *S. aureus*, MRSA, *P. aeruginosa* and *C. acnes* with the percentage of inhibition at 27.30–79.78%. Moreover, SCP at 12 h was able to inhibit bacteria by 28.93–68.34% (Table 5). Hence, SCP at 9 h and SCP at 12 h showed the highest inhibition of DNA synthesis in MRSA, with values of 79.78% and 68.34%, respectively. Similar to the phytochemical compounds, gallic acid and quercetin also reduced DNA synthesis in MRSA with the highest percentage of inhibition at 89.53% and 83.43%, respectively. Moreover, SCP at 9 h showed a significantly higher inhibition of DNA synthesis in *S. aureus* and MRSA than SCP at 12 h. In contrast, SCP at 12 h showed a significantly higher inhibition of DNA synthesis in *P. aeruginosa* and *C. acnes* than SCP at 9 h (Table 5).

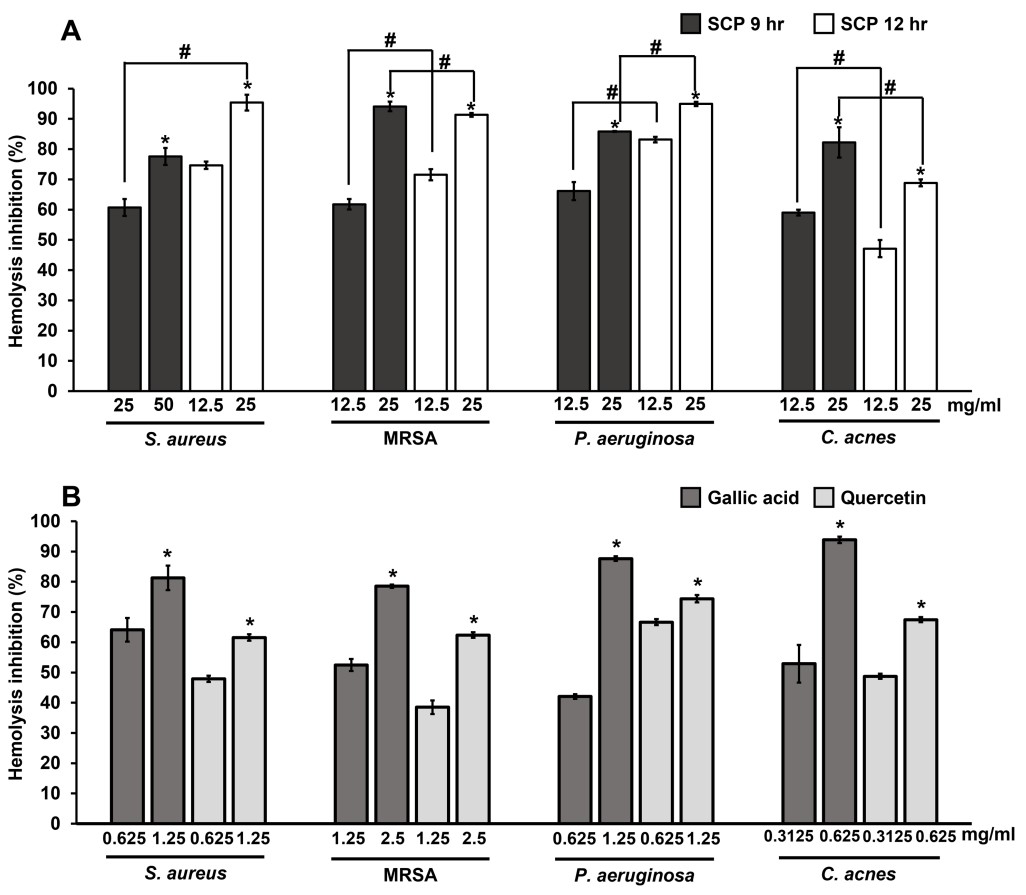

**Figure 5** **Effect of silk cocoon with pupae extracted by boiling for 9 h (SCP 9 h) and 12 h (SCP 12 h) (A) and phytochemical compounds of gallic acid and quercetin (B) on the hemolysis inhibition of *S. aureus*, MRSA, *P. aeruginosa* and *C. acnes*.** The average percentage of hemolysin inhibition of bacteria is presented as mean $\pm$ SD of three independent experiments. *Values show the highest significant hemolysin inhibition ($P < 0.05$). #Hemolysis inhibition shows significantly different values between SCP 9 h and SCP 12 h ($P < 0.05$).

## DISCUSSION

Skin infections occur when microbes invade the skin and the underlying soft tissues since pathogenic bacteria are able to adhere, multiply and invade the host. The most common skin infection is *S. aureus*, which accounts for up to 75% of cases, including patients infected with MRSA (*Singer & Talan, 2014*). Other Gram-positive bacteria, notably *C. acnes*, are prevalent anaerobic pleomorphic bacilli, showing a preference for thriving in the sebaceous glands of the skin. *C. acnes* has the capacity to generate free fatty acids within the sebaceous gland, which may irritate the follicular wall, triggering inflammation and contributing to cutaneous infections (*MacLeod, Cogen & Gallo, 2009*). They can cause dermal infections such as carbunculosis, cellulitis, erysipelas, ecthyma, impetigo, necrotizing fasciitis, folliculitis and furunculosis(*Chiller, Selkin & Murakawa, 2001*). Moreover, *P. aeruginosa*, well-known Gram-negative bacteria, may cause cutaneous infections (*Szliter-Berger &*

**Table 5  Effect of silk extracts and phytochemical compounds on the inhibition of bacterial DNA synthesis.**

| Samples | Dose (mg/mL) | DNA synthesis inhibition of bacteria (%) | | | |
|---|---|---|---|---|---|
| | | *S. aureus* | MRSA | *P. aeruginosa* | *C. acnes* |
| Silk extracts | | | | | |
| SCP 9 h | 25 | 65.98 ± 0.76[a] | 79.78 ± 1.12[a,*] | 27.30 ± 0.23[a] | 57.61 ± 1.54[a] |
| SCP 12 h | 25 | 30.25 ± 1.93[b] | 68.34 ± 0.65[b,*] | 28.93 ± 0.13[b] | 65.22 ± 0.23[b] |
| Phytochemical compounds | | | | | |
| Gallic acid | 0.625 | 63.30 ± 2.58 | 89.53 ± 0.28[*] | 78.49 ± 0.58 | 61.68 ± 0.77 |
| Quercetin | 0.625 | 61.60 ± 0.18 | 83.43 ± 0.47[*] | 82.02 ± 0.63 | 69.88 ± 1.39 |

**Notes.**
The data of different superscript letters (a, b) are presented as mean ± SD of triplicate independent experiments and show significantly different values in each silk extract ($P < 0.05$). All data are used to analyze between two groups using one-way ANOVA and Tukey's test for multiple comparisons.
Abbreviations of silk cocoon with pupae extracts from boiling for 9 h (SCP 9 h) and 12 h (SCP 12 h).
*Values show the highest significant of each sample ($P < 0.05$).

*Hazlett, 2010*). Among skin infections, there has been a rise in more resistant bacteria, with a notable increase in *S. aureus* infection by 45.9%. Among these cases, 40% were identified as MRSA infection (*Rennie, Jones & Mutnick, 2003*). Additionally, *P. aeruginosa, Enterococcus* species and beta-hemolytic streptococci infections accounted for 10.8%, 8.2%, and 2.3%, respectively as reported by hospitals in North America (*Rennie, Jones & Mutnick, 2003*). In a US population-based study, the proportion of MRSA infections increased steadily from 13% in 1998 to 48% in 2009 (*Sutter et al., 2016*; *Acree, Morgan & David, 2017*). Currently, antibiotics are the standard therapeutic treatment for skin infections, as they limit the impact on other microorganisms inhabiting the skin surface. Natural products increasingly play a pivotal role in the development of various strategies for treating diseases. Their diverse bioactive compounds often contribute to the exploration of innovative therapeutic approaches.

In this study, extracts from silk cocoon processing were assessed as potential sources of biologically active compounds for the management of human skin infections. The SP and SCP extracts contained similar quantities of phenolic and flavonoid compounds, and both demonstrated high antibacterial and antioxidant activity. The probable explanation of SP containing a significant amount of phenolics and flavonoids is that they are associated with the components of mulberry leaves, as they are the standard diet of *B. mori* (*Yasumori et al., 2002*). Longer extraction time reduces the number of phenolic compounds, which is clearly exhibited in the values of SCP extracted for 9 h (SCP at 9 h) and 12 h (SCP at 12 h). A similar study of *Bungthong & Siriamornpun (2021)* revealed that *B. mori* (Nangsew strains) silk cocoon extracts boiled for 6 h contained the highest amount of phenolics and flavonoids, which was greater than the extraction for 8 h. Moreover, the gallic acid was noted as a major active compound in the silk cocoon and silkworm pupae extracts. Furthermore, this study also discovered quercetin as one of the flavonoid compounds in silk cocoon without pupae. These results supported the study by *Tamura et al. (2002)* that found the flavonoid glucosides, namely quercetin 5-glucoside, quercetin 5,40-diglucoside and

quercetin 5,7, 4′-triglucoside in the yellow-green cocoon shell of a race Multi-Bi. However, these compounds were not detected in mulberry leaves, but were modified from the diet of silkworms by glucosyltransferase digestion before transferring the glucose residue to the C-5 hydroxy position of quercetin and accumulating in the silk fiber (*Hirayama et al., 2008*). The yellow color of *B. mori* cocoons, identified as flavonoid-related compounds such as c-prolinylquercetins (*Hirayama et al., 2006*), quercetin and kaempferol (*Zhao & Zhang, 2016*). Moreover, the flavonoids of *B. mori* cocoons identified by HPLC-ESI-MS, were presented as three quercetin glycosides; quercetin 5-O-beta-D-glucoside, quercetin 7-O-beta-D-glucoside, and quercetin 4′-O-beta-D-glucoside, and two kaempferol glycosides; kaempferol 5-O-beta-D-glucoside and kaempferol 7-O-beta-D-glucoside (*Kurioka & Yamazaki, 2002*). The metabolite compounds in silk cocoons were also detected by GC-MS analysis, which reveled 45 metabolites including organic acids, fatty acids, carbohydrates, amino acids, and hydrocarbons (*Zhang et al., 2017*). Therefore, the antioxidant properties of silk cocoon and silkworm pupae extracts can be attributed to the presence of phenolic and flavonoid compounds. Antioxidant substances, as highlighted by *Preedy (2012)*, possess the capability to bind and neutralize reactive oxygen species, thereby preventing oxidative damage to cells and tissues. In addition, other compounds in silk cocoon extract, such as sericin and fibroin, should be evaluated in the future for their antibacterial, anti-inflammation and wound healing abilities.

The efficacy of silk cocoon and silkworm extracts on the inhibition of skin pathogenic bacteria were evaluated, and the results demonstrated that the extracts of SCP, SCWP and SP had the ability to inhibit skin pathogenic bacteria, including both Gram-positive bacteria (*S. aureus*, MRSA, *C. acnes*), and Gram-negative bacteria (*P. aeruginosa*). SCP exhibited strong inhibitory activity against MRSA, *C. acnes*, and *P. aeruginosa*. Additionally, SP enhanced the antibacterial activity of SCP. Regarding SCWP, its inhibitory activity was greatest against *C. acnes*. In this study, the phytochemical compounds of gallic acid and quercetin exhibited inhibitory effects against all tested bacteria. In addition, the quantities of gallic acid were present in high amounts in SCP and SP, but low amounts in SCWP. However, it is noted that SCWP gained effectiveness from the quercetin compound. The antimicrobial properties of silk extracts might be attributed to the level of polyphenol compounds, especially quercetin and gallic acid (*Criste et al., 2020*). However, alternative methods should be explored to enhance the concentration of quercetin and gallic acid in the extracts. This could involve optimizing the extraction process, modifying extraction conditions, or considering additional purification steps in future studies.

Several published studies have reported the antibacterial activity of *B. mori* silk to be present in the proteins of sericin and seroin. Sericin extracted from the cocoons of *B. mori* demonstrated a reduction of bacterial growth by 89.4% and 81% against *S. aureus* and *Escherichia coli*, respectively (*Rajendran et al., 2012*). According to the findings of *Singh et al. (2014)*, seroin-2 effectively hindered the growth of *E. coli* and *Micrococcus luteus*, whereas seroin-1 exclusively inhibited the growth of *M. luteus*.

SCP at 9 h and SCP at 12 h had stronger antibacterial activity against skin pathogenic bacteria than SCWP. Furthermore, the extracts showed bactericidal activity by reducing the viability of cells within 2-10 h. We have speculated the antimicrobial activities, which

may involve various mechanisms, namely on preformed and mature biofilms, hemolysis activity and DNA synthesis of skin pathogenic bacteria. Our recent research showed a new report of the inhibition effects of silk cocoon extracts on skin pathogenic bacteria. We also demonstrated that SCP at 12 h at a concentration of 25 mg/mL could inhibit preformed and mature biofilms of bacteria by more than 50%. SCP extracted at 9 h was especially able to reduce preformed and mature biofilms of antibiotic-resistant bacteria, namely MRSA, by more than 87%. Hence, gallic acid and quercetin also contributed to the inhibition of preformed and mature biofilm on all tested skin pathogenic bacteria. Biofilm-linked pathogens are persistent biofilms that are aggregates of microorganisms, and form a multilayered, heterogeneous, microbial mat for the attachment mechanisms of bacterial infections, and also protect the immune defenses (*Flemming & Wuertz, 2019*). Moreover, various treatments have low efficiency and high toxicity while treating biofilm formation on bacteria (*Mishra et al., 2020*). Recent research on gallic acid revealed strong antibiofilm activity against *S. aureus* (*Kang et al., 2008*), *Streptococcus mutans* and *E. coli* (*Shao et al., 2015*). Quercetin also inhibited biofilm formation in *S. aureus* by the downregulation of *icaA* and *icaD* genes, which are encoded genes for biofilm production (*Lee et al., 2013*).

In fact, most skin pathogenic bacteria produce hemolysin, which has cytotoxic, dermonecrotic and hemolytic effects (*Dinges, Orwin & Schlievert, 2000*). Moreover, the inhibition of hemolysis is an accepted as anti-virulence factor (*Wyatt et al., 2010*). The present study showed, for the first time, that SCP extracts were able to prevent blood hemolysis from skin pathogenic bacterial infections by more than 60%. In addition, the phytochemical compounds of gallic acid and quercetin were important non-sericin compounds in silk cocoon extracts and strongly inhibited blood hemolysis activity of skin pathogenic bacteria. In previous studies, other non-sericin including flavonoid compounds; such as flavone, quercetin, apigenin and fisetin, decreased blood hemolysis that induced by *S. aureus* (*Lee et al., 2012*). Similarly, gallic acid was able to inhibit alpha-hemolysin produced by *S. aureus* with hemolytic, cytotoxic, dermonecrotic and lethal properties (*Luís et al., 2014*). In other skin pathogenic bacteria, hemolytic activity occurring in *P. aeruginosa* was inhibited by various phytochemical compounds such as eugenol, cinnamaldehyde and resveratrol (*Lee et al., 2014*; *Kim et al., 2015*).

Therefore, the potential of SCP extracts to inhibit biofilm formation and prevent hemolysis have shown a way to reduce bacterial colonization and invasion on epithelial host cells, which subsequently leads to infections. Hence, this study also assessed the ability of SCP extracts to inhibit the DNA synthesis of bacteria. Moreover, SCP extracts exhibited the highest inhibition on DNA synthesis in MRSA. Positive results also correlated to the reduction of bacteria in DNA synthesis by gallic acid and quercetin compounds. Kaempferol and quercetin have been proposed as antibacterial agents due to their interaction with two crucial bacterial enzymes; DNA gyrase, and DNA topoisomerase IV. DNA gyrase plays a role in controlling DNA supercoiling and alleviating topological stress associated with the translocation of transcription and replication (*Zhang et al., 2008*). Polyphenols could impact the inhibition of cell wall formation and cytoplasmic membrane disruption, and interfere with nucleic acid synthesis (*Makarewicz et al., 2021*).

## CONCLUSION

Our study revealed the biological activity of SCP and SCWP, and also involved SP in order to observe the inhibition of skin pathogenic bacteria such as *S. aureus*, MRSA, *P. aeruginosa* and *C. acnes*. SCP was proved to be effective against all tested bacteria, and SP also contributed to inhibitory activity. In addition, silk cocoon extracts and silkworm pupae were found to contain phenolics, flavonoids and antioxidant properties. Hence, a high amount of gallic acid obtained from SCP and SP also supported antibacterial activity. However, SCWP contained phytochemical compounds of gallic acid and quercetin, which had the ability to inhibit bacteria. Their bioavailability from SCP and phytochemical compounds strongly contributed to the inhibition of preformed and mature biofilms, blood hemolysis and DNA synthesis of skin pathogenic bacteria, which were emphasized as a new report from this study. This finding suggests additional benefits of silk cocoon extract for the prevention of skin pathogenic bacterial infections, alongside its antioxidant activity. Therefore, the natural byproduct from silk processing has demonstrated valuable biological activity that can be applied as therapeutic treatment for skin infections.

## ACKNOWLEDGEMENTS

We would like to express our thanks to Julia Akins for language editing.

### Funding

This research was financially supported by the Thailand Research Fund, Research and Researcher for Industry (RRi) Master Scholarship (MSD58I0021) and the Natural Extracts and Innovative Products for Alternative Healthcare Research Group, Chiang Mai University Chiang Mai, Thailand. There was no additional external funding received for this study. The funders had no role in study design, data collection and analysis, decision to publish, or preparation of the manuscript.

### Grant Disclosures

The following grant information was disclosed by the authors:
Thailand Research Fund, Research and Researcher for Industry (RRi) Master Scholarship: MSD58I0021.
Natural Extracts and Innovative Products for Alternative Healthcare Research Group, Chiang Mai University Chiang Mai, Thailand.

### Competing Interests

The authors declare there are no competing interests.

### Author Contributions

- Thida Kaewkod performed the experiments, analyzed the data, prepared figures and/or tables, authored or reviewed drafts of the article, and approved the final draft.

- Puangphaka Kumseewai performed the experiments, analyzed the data, authored or reviewed drafts of the article, and approved the final draft.
- Sureeporn Suriyaprom performed the experiments, authored or reviewed drafts of the article, and approved the final draft.
- Varachaya Intachaisri performed the experiments, authored or reviewed drafts of the article, and approved the final draft.
- Nitsanat Cheepchirasuk performed the experiments, authored or reviewed drafts of the article, and approved the final draft.
- Yingmanee Tragoolpua conceived and designed the experiments, authored or reviewed drafts of the article, and approved the final draft.

## Data Availability

The raw data is available in the Supplemental Files.

## Supplemental Information

Supplemental information for this article can be found online at http://dx.doi.org/10.7717/peerj.17490#supplemental-information.

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
