# Peer review of "Potential therapeutic agents of Bombyx mori silk cocoon extracts from agricultural product for inhibition of skin pathogenic bacteria and free radicals"

_PeerJ, doi:10.7717/peerj.17490_

## Round 0.1 · original submission · Minor Revisions

Both reviewers have identified areas within that manuscript that should be improved.

With regard to the second comment from Reviewer 1, while it may not be feasible for the authors to undertake an MS analysis of the samples I think they could provide additional information on any other compounds identified by HPLC.

In addition to the comments from the reviewers, I note the following points that should be addressed:

The use of English must be improved. There are numerous grammatical and typographical errors throughout, and in some cases this makes the article difficult to understand. For example:

* Line 21: "Antioxidant activity of silk cocoon extracts was demonstrated." should read: "Antioxidant activity of silk cocoon extracts has been reported."
* Throughout, the parentheses "()" and "%" signs appear to be either a different font size or in a different font. These should be corrected.
* "Folin–Cioculteu" should be spelt "Folin–Ciocalteu"
* Line 81: "It helps protect physical impacts and prevent dehydration" should read: "It helps protect against physical impacts and prevent dehydration."
* Line 426: "In this study, the extract from silk cocoon processing is an alternative interest in biological substances for the management of human skin infections." This sentence doesn't make sense. It should be something like: "In this study, extracts from silk cocoon processing represent a potential source of alternative biologically active compounds for the management of human skin infections."

These are just a few of many errors. I strongly encourage the authors to have their manuscript reviewed and corrected by a professional English language editing service.

Further comments:
Line 56: The authors should define sericulture (i.e., "the process of cultivating silkworms and extracting silk"). This term may not be familiar to all readers.

Line 56/57: "Thai sericulture was established a long time ago." This is a vague statement. Could the authors provide more detail (was this hundreds of years ago? Thousands?) and/or cite an appropriate reference?

Line 109: "Consequently, harnessing the extracts from silk cocoon and silkworm..." I suggest that the authors first mention the more general idea of bioactive compounds, then specifically refer to silk extracts. For example: "Consequently, exploring bioactive compounds, including extracts from silk cocoon and silkworm, as beneficial alternatives providing safe and effective treatment of human skin infections is an active area of research."

Line 120: The authors should provide an explanation of why the time periods chosen for boiling the various extracts were so variable (i.e., 1 h for the pupae, up to 12 h for cocoon with pupae), or provide a relevant reference for using these times. Had they done previous experiments to establish that these were suitable times for preparing the samples? [Note, a similar point is raised by Reviewer 1].

Results: When the authors mention a value being higher (for example, line 283) they should provide the p-value to support this observation.

Discussion:
Line 417: "Among these cases, 40% were identified as MRSA." Please cite an appropriate reference for this statement or, if it is citing the Rennie et al 2003 paper, please make this clear.

Lines 427 to 433: These lines are repeating methods and results and are not required. Please also check elsewhere in the discussion to ensure methods and results are not re-stated.

**Language Note:** The Academic Editor has identified that the English language must be improved. PeerJ can provide language editing services - please contact us at [email protected] for pricing (be sure to provide your manuscript number and title). Alternatively, you should make your own arrangements to improve the language quality and provide details in your response letter. – PeerJ Staff

Reviewer 1 ·

Basic reporting

This study focused on the antimicrobial components in the silkworm cocoon, but the authors did not mention the presence of non-sericin antimicrobial proteins in the silk, including seroins and protease inhibitors. In addition to flavonoids and carotenoids, there are also various nonprotein components in silkworm cocoons, including organic acids, fatty acids, carbohydrates, amino acids, and hydrocarbons. I suggest that the authors should provide more comprehensive literatures on silk components to be useful to readers.

Experimental design

no comment

Validity of the findings

How do the authors determine the best extraction time?

What are the substances identified by HPLC in addition to quercetin and gallic acid? It is recommended to use the mass spectrometry to identify the total components.

Since that SP has the most abundant phenolics and flavonoids, and SCWP contains the most abundant quercetin, how to explain why SCP has the best antibacterial activity? Does this mean that antibacterial activity is not primarily due to phenolics and flavonoids?

The MIC and MBC of the cocoon extracts are too high. The authors can try a better way to increase the concentration of quercetin and gallic acid in the cocoon extracts.

·

Basic reporting

Literature reports are sufficently quoted and the overall structure of the manuscript is systematically represented. Additional comments are listed in below section.

Experimental design

Experimental design is systematically represented. Additional comments are listed in below section.

Validity of the findings

Authors have significant findings on the extracts of silk cocoon and silkworm for their antioxidant activity and phytochemical moieties. The experiments were carefully carried out and finidngs are significant.

Additional comments

1.In HPLC chromatogram of Gallic acid standard showed a pre-peak. How is the intergration done? Integrated chromatogram should be provided in the manuscript for reader to closely look at.
2. In table 4, it is good to represent Gallic acid and Quercetin with unifrom significant digits

---

## Round 0.2 · Minor Revisions

The authors have tried to address the comments from the two reviewers, as well as my own comments, and as a result, the manuscript shows clear improvement. However, there further improvement is needed before it is suitable for publication.

Specifically:

1. The authors still include information on both methods and results within the discussion section. Lines 436 to 444 could be considerably shortened, for example: "In this study, extracts from silk cocoon processing were assessed as potential sources of biologically active compounds for the management of human skin infections. The SP and SCP extracts contained similar quantities of phenolic and flavonoid compounds, and both demonstrated high antibacterial activity." Then the authors can go on to discuss the factors that might influence the phenolic / flavonoid content and antibacterial activity.
Similarly, lines 453 to 457 and lines 482 to 487 contain information that has already been reported in the results section. The authors need to check that any descriptions of methods or results in the discussion are brief and are necessary for the point being discussed.

2. The use of English has improved, however, it is still not of sufficient quality for publication. This is clear from the opening sentence of the abstract: "Most skin diseases are caused by pathogenic bacteria that drive the research to discover new therapeutics for treatment." This suggests that the pathogenic bacteria are driving research to discover new therapeutics. However, as the authors later state in the introduction, it is actually "issues like resistance and environmental degradation" associated with current therapeutic options that are driving the research for alternatives.
I once again strongly recommend that the authors have their manuscript reviewed by a professional English language editing service (note that PeerJ provides such a service).

3. The authors have revised Table 4 as requested by reviewer 2, however further work is required. In the Table legend the authors state that "The results are presented as mean +/- SD" however no SD data are included in the table. They also have superscript "a" used throughout the table - this is not necessary because there are only 4 values with a different superscript. I suggest the authors remove the "a" superscript and replace "b" and "c" with symbols to show differences. It is also not clear what differences are being shown. Are they showing a different effect between the extracts within a bacterial strain, different effects between bacterial strains within a particular extract, or both? This needs to be made clear when the different superscript symbols are defined.

**Language Note:** The Academic Editor has identified that the English language must be improved. PeerJ can provide language editing services - please contact us at [email protected] for pricing (be sure to provide your manuscript number and title). Alternatively, you should make your own arrangements to improve the language quality and provide details in your response letter. – PeerJ Staff

---

## Round 0.3 · Minor Revisions

This version of the manuscript is a further improvement, however the discussion and conclusion are still in parts difficult to follow and would benefit from further editing. I recommend that these sections undergo one final revision for quality of language and clarity.

---

## Round 0.4 · accepted · Accept

The authors' revisions of the discussion and conclusion have further improved the readability and clarity of the paper. I believe it is now acceptable for publication.